# Improving CNV Detection Performance in Microarray Data Using a Machine Learning-Based Approach

**DOI:** 10.3390/diagnostics14010084

**Published:** 2023-12-29

**Authors:** Chul Jun Goh, Hyuk-Jung Kwon, Yoonhee Kim, Seunghee Jung, Jiwoo Park, Isaac Kise Lee, Bo-Ram Park, Myeong-Ji Kim, Min-Jeong Kim, Min-Seob Lee

**Affiliations:** 1Eone-Diagnomics Genome Center, Inc., 143, Gaetbeol-ro, Yeonsu-gu, Incheon 21999, Republic of Korea; cj.ko@edgc.com (C.J.G.); hjkwon@edgc.com (H.-J.K.); yh.kim@edgc.com (Y.K.); sh.jung@edgc.com (S.J.); jw.park@edgc.com (J.P.); ks.lee@edgc.com (I.K.L.); br.park@edgc.com (B.-R.P.); myeongji.kim@edgc.com (M.-J.K.); 2Department of Computer Science and Engineering, Incheon National University (INU), Incheon 22012, Republic of Korea; 3NGENI Foundation, San Diego, CA 92127, USA; 4Diagnomics, Inc., 5795 Kearny Villa Rd., San Diego, CA 92123, USA; mjkim@diagnomics.com

**Keywords:** CNV, genome-wide SNP array, Korean newborn, machine learning, genomic wave

## Abstract

Copy number variation (CNV) is a primary source of structural variation in the human genome, leading to several disorders. Therefore, analyzing neonatal CNVs is crucial for managing CNV-related chromosomal disabilities. However, genomic waves can hinder accurate CNV analysis. To mitigate the influences of the waves, we adopted a machine learning approach and developed a new method that uses a modified log R ratio instead of the commonly used log R ratio. Validation results using samples with known CNVs demonstrated the superior performance of our method. We analyzed a total of 16,046 Korean newborn samples using the new method and identified CNVs related to 39 genetic disorders were identified in 342 cases. The most frequently detected CNV-related disorder was Joubert syndrome 4. The accuracy of our method was further confirmed by analyzing a subset of the detected results using NGS and comparing them with our results. The utilization of a genome-wide single nucleotide polymorphism array with wave offset was shown to be a powerful method for identifying CNVs in neonatal cases. The accurate screening and the ability to identify various disease susceptibilities offered by our new method could facilitate the identification of CNV-associated chromosomal disease etiologies.

## 1. Introduction

Developmental disabilities can impact a range of domains, including perception, cognition, movement, and language. The disabilities predominantly arise from chromosomal abnormalities, such as copy number variations (CNVs). CNVs refer to large deletions or duplications of genomic material that are greater than 1 kilobase (kb) in size [1]. Although most CNVs are functionally benign, they are a common source of genomic structural variation [2,3,4], and some of the variations are associated with various diseases, such as intellectual disability, autism, schizophrenia, and developmental disorders [5,6,7,8,9]. Therefore, early and accurate detection of CNVs is essential for providing appropriate interventions and support to individuals and families affected by the CNVs.

Numerous methods exist for detecting chromosomal abnormalities, encompassing conventional techniques such as karyotyping, fluorescence in situ hybridization (FISH) [10], and multiplex ligation-dependent probe amplification [11], as well as contemporary approaches like chromosomal microarray analysis (CMA) [12]. These tests can be used to diagnose genetic disorders, including Down syndrome, Turner syndrome, and some forms of cancer. In particular, chromosomal tests are also applicable for carrier screening, prenatal testing, and newborn screening [13,14].

Similar to whole genome sequencing (WGS) analysis, CMA is a high-resolution technique to screen the entire genome and identify CNVs [15]. While WGS can identify CNVs, SNPs, and other genetic variations, providing all-encompassing information of the entire genome, the cost and time requirements of WGS surpass those of microarray analysis, making it less feasible for regular clinical testing purposes [16,17]. Using an array with probes designed to selectively bind with DNA extracted from a sample, CMA demonstrates the capability to detect CNVs as small as 50–100 kb in size. Notably, this ability enables CMA to detect difficult-to-identify diseases, including developmental disorders and multiple congenital anomalies, with a detection rate of 15–20% [18,19,20].

CMA allows for the simultaneous detection of both CNVs and rare mutations in a single run [21]. For example, the Illumina Infinium Global Screening Array (GSA) is able to scan approximately 750,000 SNPs across the entire human genome [22]. By utilizing the log R ratio (LRR), a normalized signal intensity value for individual SNP markers, and B allele frequency (BAF), a normalized allelic intensity value for two alleles, data from the microarray, CNVs, and rare mutations can be detected [23]. This approach facilitates comprehensive genetic screening and analysis across diverse populations.

During the analysis of CNVs using microarrays, the presence of wave-like patterns characterized by genome-wide spatial autocorrelation has been noted [24,25,26]. The patterns were observed at the chromosomal level rather than in narrow subregions. Moreover, those were evident even when copy numbers were normal, attributed to the high variability of LRR. This phenomenon, referred to as a genomic wave, is speculated to be caused by variations in both quantity and quality of DNA. The pattern, observed across all chromosomes and varying between samples, is known to have negative impacts on the accuracy of CNV detection [24].

Since the identification of genomic waves, various methods have been developed and utilized to improve the accuracy of CNV detection in the presence of these waves. These methods include the utilization of Loess [27] and the Genomic Imbalance Map algorithm [28], in addition to the correlation with the guanine-cytosine content of the genome sequence [24]. These strategies serve to alleviate the impact of genomic waves on CNV detection.

In this study, a new method using machine learning models was employed to mitigate the effect of the genomic waves on CNV analysis. Among the different machine learning methods available, k-means [29,30] and k-nearest neighbor (k-NN) [31,32] were selected due to their simplicity and strong performance. Using the approaches, we obtained a new LRR value called modified LRR (mLRR). The effectiveness of the new method on CNV analysis was validated by comparing the results of analyzing samples with known CNVs, and the results from next-generation sequencing (NGS) were utilized to confirm the accuracy of our method. As a result of the validation, the new method showed a greater performance than the original one.

## 2. Materials and Methods

### 2.1. Subjects and Sample Preparation

This study was performed in accordance with the 2021 Guidelines for using health data by the Ministry of Health and Welfare of Korea. We analyzed the DNA CNVs in 16,046 peripheral or cord blood samples collected from newborn Korean babies. Each blood sample (0.1 mL) was placed into a BD Microtainer tube with K2EDTA (BD, Franklin Lakes, NJ, USA) and analyzed at clinical centers for genetic analysis between February 2018 and May 2021. The blood samples were transported at room temperature to the laboratory, where genomic DNA was extracted from the blood using a Chemagic DNA Blood 200 Kit (Perkin Elmer, Waltham, MA, USA) according to the manufacturer’s protocol. Before performing the microarray assay, the genomic DNA concentration and purity were measured using an EpochTM microplate spectrophotometer (BioTek, Winooski, VT, USA). Genomic DNA (200 ng) was used to generate targets for the Illumina Infinium HTS assay protocol, and NGS was performed using 500 ng of genomic DNA.

### 2.2. Subjects and Sample Preparation

We custom-engineered the Illumina Infinium GSA BeadChip (version 2, Illumina, San Diego, CA, USA) to include 742,759 SNP markers capable of detecting 138 CNV-related chromosomal disorders (Figure 1). Procedures for DNA amplification, fragmentation, hybridization, and staining were performed according to the Illumina Infinium HTS assay protocol (Illumina). Image and data files were obtained using the iScan control and Genome Studio software packages (v2.0.4) from Illumina. 

### 2.3. Preparation of Positive CNV Control Samples for Analytical Validation

The SNP array analysis was validated using 22 human cell line DNA samples provided by the Coriell Institute for Medical Research. Each analysis was repeated 2 or 3 times for reproducibility and accuracy. All experimental methods used in association with the human cell lines were performed in the same manner as those performed for newborn specimens. 

### 2.4. Data Processing and CNV Analysis

Raw data from each sample were processed using in-house tools to generate the signal intensities (expressed as LRR) and allelic intensity ratios (expressed as BAF) of all SNPs. To ensure data quality, only markers with call rates of ≥0.98 and LRR SDs of ≤0.2 were selected.

The PennCNV [33] and QuantiSNP [34] were performed to identify copy number deletions and duplications using population frequencies of the B allele, which were calculated based on the BAF of each marker in 1100 samples. Subsequently, adjacent CNVs that were <200 kb apart were merged and filtered out based on the SNP number (>10), CNV length (>50 kb), and confidence score (>50), all of which were generated using PennCNV and QuaintiSNP. The CNVs detected by each program were compared against our custom database, which contains positional data related to 138 chromosomal disorders associated with CNVs. Following this comparison, only the CNVs that corresponded to each specific disease were selected for further analysis. The results obtained from the programs were merged to reduce false negatives. The final result of the analysis was either the detection of a CNV (if the condition was met) or normal status (if the condition was not met) (Figure 2). 

### 2.5. Clustering Genomic Waves from 5399 Clinical Samples Using GSA

The genomic waves in the results of the GSA chip were identified using 5399 clinical samples. Autosomal chromosomes were divided into 1 Mb bins, and the LRR means, and SDs of markers within the region were calculated. Bins with no markers or LRR SDs of ≤0.05 were excluded from the analysis because small SDs result in even distribution and are not useful for analysis. As such, 238 domains were created, and the LRR mean was used as the feature for analysis.

To cluster the waves into patterns, it was necessary to determine the optimal number of clusters. This was achieved by calculating the sum of distances between the cluster center and its members while incrementally increasing the number of clusters from 2 to 20. The ‘elbow point’ of the value, as determined using the elbow method [35], indicated that the decrease of k-means becomes smaller after 6 clusters. As a result, 6 was chosen as the optimal number of clusters. 

In total, 5399 samples were clustered using the k-means method with the optimal value of 6. The 6 clusters represented distinct wave patterns and consisted of 768, 1202, 1241, 743, 788, and 657 samples, respectively. To determine the clustered LRR pattern, each sample was divided into 1 Mb portions, and the mean LRR of the included markers was calculated. For samples in the same cluster, the mean LRR mean was calculated between them. The resulting clustered data were subjected to dimension reduction analysis methods, specifically t-distributed stochastic neighbor embedding and principal component analysis. Subsequent plots and comparisons with the k-means clusters were conducted. In both analyses, the samples were effectively categorized into the 6 clusters (Figure 3).

### 2.6. Cluster Matching of Analytical Samples and Calculation of Modified LRR Values

The LRR mean of 238 regions used for k-means analysis was calculated and classified using k-NN. The LRR data for matched cluster samples were normalized into Z-scores using the following formula:$$\mathrm{Zi}=\frac{\mathrm{Xi}-\bar{X}i\;}{\mathrm{Si}\;}$$
where X represents the LRR value in the sample, $\bar{X}$ represents the mean, and S represents the standard deviation (SD) calculated for the samples within the group.

Due to the differences in the range of normalized values compared to the original LRR, adjustment to the original range was required. This involved resizing the original LRR SD and Z-score SD to a similar value, resulting in the creation of a new LRR value referred to as the modified LRR (mLRR). The offset effect was verified at the chromosome level: a wave was observed in the results using the LRR, while no wave was observed in the results of the mLRR (Figure 4a). This phenomenon was particularly pronounced at each end of the chromosomes (Figure 4b).

### 2.7. NGS Sequencing for Accuracy Validation

To confirm the accuracy of our method using NGS, all genomic DNA passing our QC criteria (OD260/OD280 ≥ 1.8; 1.9 ≤ OD260/OD230 ≤ 2.2) were prepared for library construction. Briefly, 30 ng of genomic DNA was sheared into small fragments (170–200 bp) using a focused M220 ultrasonicator (Covaris, Woburn, MA, USA). Following end repair, the addition of an A overhang, and adapter ligation, all ligated fragments were cleaned up using Hiaccubead magnetic beads (Accugene, Incheon, Korea). Libraries were prepared using the Accel-NGS 2S Plus DNA Library Kit (Swift Biosciences, Ann Arbor, MI, USA) according to the manufacturer’s protocols. The size distribution of each library was assessed using a 4200 Tapestation system (Agilent Technologies, Palo Alto, CA, USA). The libraries were sequenced using an Illumina NextSeq platform with paired-end sequencing (36 × 2) following the manufacturer’s protocols.

The Ion Torrent Proton platform from Thermo Fisher Scientific was also used as follows. Libraries were prepared using an Ion AmpliSeq Library Kit 2.0 (Thermo Fisher Scientific, Waltham, MA, USA). Adapter ligation, end repair, PCR amplification, and barcoding were performed using an Ion Xpress Adapter 1–96 Kit (Thermo Fisher Scientific). An Ion Chef system was used to complete emulsion PCR and enrichment steps according to the manufacturer’s protocol. The resulting libraries were sequenced using an Ion Torrent Proton system with an Ion PI Chip Kit V3 (Thermo Fisher Scientific).

The sequencing data were aligned to the hg 19 human reference genome using Burrows-Wheeler Aligner (ver. 0.7.15) [36]. Using in-house software, duplicated reads were removed, and read depths and z-scores for each position were calculated.

## 3. Results

### 3.1. Enhancing CNV Analysis Accuracy through Customized Machine Learning Model

To address the issue of wave patterns that can impede accurate CNV analysis using an array, we developed a customized machine-learning analysis. This involved three processes: (1) clustering wave patterns with k-means, (2) classifying samples into their nearest cluster using k-NN, and (3) utilizing the original LRR standard deviation (SD), mean values, and Z-score SD values. These approaches enabled us to normalize and offset waves, thereby improving the accuracy of the array analysis. Modified log R ratio (mLRR) values, which would serve as input parameters for the CNV analysis tool, were obtained.

### 3.2. Analytical Validation Using Known Positive CNV Control DNA Samples

A total of 22 samples from the Coriell cell line repository with defined chromosomal abnormalities were analyzed using LRR and mLRR values to assess performance for the detection of CNVs between the two values. To confirm reproducibility and evaluate the accurate performance, the analysis was repeated 2 or 3 times for each sample (Figure 5 and Appendix A).

As a result, the utilization of mLRR yielded a more powerful detection performance than the original one. All the known CNV regions were detected with our new method from the 67 analyses, whereas some regions were missed in the analysis based on the standard LRR values (Figure 5 and Appendix A). Among the 67 repeats, 7 (10.45%) were not detected in the standard LRR analysis. The length of the detected CNV was higher or the same as that of mLRR in all analyses, except for cases where it was not detected. For example, in the case of GM08039 with a known CNV spanning 22,723,028 bp associated with Trisomy 16, the mLRR method detected 99.977%, 95.508%, and 99.977% of the CNV region in three repeats, respectively. In contrast, when LRR was employed, CNV was not detected in 2 analyses out of 3 repeats, and only 0.404% was detected in one case. In the analysis of GM05876, which harbors a 1,435,491 bp CNV associated with DiGeorge syndrome, the CNV was detected in all analyses using LRR and mLRR. However, when LRR was employed, only 8.792%, 46.376%, and 24.800% were detected in three repeated analyses, respectively. In contrast, the method using mLRR exhibited high detection rates of 83.205%, 83.205%, and 99.843% (Figure 5 and Appendix A).

### 3.3. CNV Analysis Using 16,046 Neonate Samples from South Korea

From February 2018 to May 2021, we collected 16,046 neonate samples from the clinical centers located in South Korea. We utilized the mLRR values, whose performance had been validated, to analyze the samples and attempted to detect 138 CNV-related chromosomal disorders (Figure 1) using a customized GSA BeadChip. 

As a result of the screening, the genome-wide SNP array chip targeting 138 CNV-related chromosomal disorders identified 342 cases of 39 CNV-associated chromosomal disorders (Figure 6 and Appendix A). 

The most frequently detected disorder was Joubert syndrome 4 in the 2q13 region (66 of 342 cases). The 2q13 microdeletion encompasses genes encoding a MAL-like protein and nephrocystin 1 (NPHP1). A homozygous deletion of NPHP1 on chromosome region 2q13 is known to cause a rare genetic disorder, Joubert syndrome 4 [37,38]. The syndrome shows a condition in which parts of the brain do not develop properly. All the 66 cases of 2q13 deletions were identified as heterogeneous deletions [arr[hg19] 2q13 (110,852,875–110,983,320) × 1]. The detected cases were presumed to be carriers of the Joubert syndrome 4.

The second most commonly detected chromosomal abnormalities (55 cases) were located in the 15q11.2 region. The 15q11.2 microdeletion pertains to a characteristic 500 kb (0.5 Mb) deleted segment situated between breakpoint 1 (BP1) and breakpoint 2 (BP2). Approximately 8–10% of the individuals with 15q11.2 deletions exhibit characteristics such as developmental delays in motor and language skills [39]. Disruptions of genes within the 15q11.2 region result in an autosomal dominant form of disability with low penetrance. It might offer a plausible explanation for the higher-than-normal frequency in the population. The prevalence of this relatively high frequency is corroborated by a study carried out by the University of Kansas Medical Center [40], which highlights that CNVs at the 15q11.2 BP1-BP2 microdeletion region are estimated to be present in 0.5% to 1.0% of the population.

The third and fourth most frequently detected CNV-related disorders were the 22q11.2 duplication syndrome and the 17p13.3 telomeric duplication syndrome, which were detected in 47 and 33 cases, respectively. The features of the 22q11.2 duplication syndrome, which is caused by an extra copy of a piece of chromosome 22 containing about 30–40 genes, are known to be varied even among family members (i.e., intrafamilial variability exists) [41]. Some with the duplicated gene exhibit intellectual or learning disabilities in addition to developmental delay, slow growth, and weak muscle tone (hypotonia) [42]. Duplications involving one or more genes on chromosome 17p13.3 are associated with split-hand/foot malformation and long-bone deficiency-3 (SHFLD3), with the duplication of the basic helix-loop-helix transcription factor of the A9 (BHLHA9) gene especially associated with limb defects. SHFLD3 is a relatively rare autosomal dominant skeletal disease with a penetration rate of <50% and features a broad spectrum of intraindividual variability [43,44]. The following genetic disorders involving CNVs were found in 11–15 of 16,046 cases: 2p16.3 deletion, DiGeorge, 1q21.1 microdeletion, and Klinefelter syndromes (Appendix A).

To verify the accuracy of our GSA array–based approach in a clinical setting, a comparison was performed with the results obtained using next-generation sequencing (NGS). From the results of the 16,046 samples, we selected CNV-associated chromosomal disorders that were frequently detected in this study, as well as those known to be rare, for comparison (Figure 7 and Appendix A). 

The same DNA samples, analyzed with the custom-engineered chip, were analyzed using NGS. Read depths for each position were computed using reads aligned to hg19, the human reference genome. Subsequently, z-scores were calculated for all positions based on these read depths. Our analysis revealed a distinctive variation in z-score within the genomic region where the GSA-identified CNV was located, distinguishing it from adjacent positions.

Comparing the NGS results with the genome-wide SNP array analysis demonstrated complete consistency, achieving a 100% match. The CNVs identified through the array analysis were precisely mirrored in the NGS findings. This underlines the high consistency and robustness of our method in accurately detecting various CNVs, showcasing its strong performance and reliability.

## 4. Discussion

CNVs are associated with many neurodevelopmental disorders, such as autism spectrum disorders, schizophrenia, intellectual disability, attention deficit hyperactivity disorder, developmental delay, and epilepsy [45,46,47]. As the variations often span several mega-base pairs that encompass multiple genes [48,49], the rarity and dose-sensitive nature of individual CNV genes must be accurately determined. For instance, altering the copy number of a dose-sensitive gene like BHLHA9 can be detrimental to disease pathogenesis, whereas changing the copy number of dose-insensitive genes is unlikely to cause harm [50]. Even within the same chromosomal region, CNVs may be associated with different phenotypes, ranges of severity, and incomplete penetration [51]. In the present study, among 39 detected cases of chromosomal abnormalities, 8 CNV-related chromosomal disorders were known to have complete penetrance: DiGeorge, Klinefelter, Down, Triple-X, Turner, Williams (duplication and deletion), and Prader–Willi/Angelman syndromes. For example, Duchenne muscular dystrophy (OMIM#310200), an X-linked recessive myopathy caused by a mutation in the dystrophin gene located at Xp21, has a penetration rate of 100% in males [52]. Charcot–Marie–Tooth syndrome type 1A (CMT1A), which was detected in four cases in our study, exhibits varying penetration rates depending on the parent. Fathers with X-linked dominant CMT1A have a 100% risk of having an affected daughter, whereas their sons face no such risk; conversely, both sons and daughters of mothers with X-linked dominant CMT1A have a 50% chance of being affected by the syndrome [53]. Joubert syndrome 4, the most frequently detected CNV-related chromosomal abnormality in our study, is a rare autosomal recessive disorder involving a ~290 kb homozygous deletion containing NPHP1 in 2q13. All Joubert syndrome 4 cases identified in our screening were heterogeneous deletions at 2q13; thus, the individuals were assumed to be carriers of the disorder in all cases. Other chromosomal disorders (i.e., 15q11.2 deletion, 22q11.2 duplication, 17p13.3 telomeric duplication, and 2p15.2 deletion syndromes, among others) detected in this study represent syndromes with various penetration rates. Additionally, while disorders involving visual and hearing impairments are often detected early, e.g., before the age of three, invisible autism, as well as emotional and behavioral disorders, are more likely to occur after the age of three [54]. Thus, it is very important to detect chromosomal abnormalities early and accurately in order to minimize the symptoms of the disease and slow its progression.

As a comprehensive and universal screening tool, the GSA method can detect various genetic abnormalities with high accuracy, making it a reliable option for large-population screening compared to other screening methods. The array offers a resolution level that is more than 10 times higher than conventional karyotyping or FISH analysis, allowing for the detection of micro-chromosomal abnormalities that are larger than 100 kb in size with higher confidence levels [55,56,57]. 

However, it is crucial to acknowledge the limitations of genotyping arrays; they are unable to detect translocations and inversions [58]. Recent studies have highlighted that low-pass genome sequencing technology surpasses microarray technology in terms of detection rate, resolution, and cost-effectiveness [59,60,61,62]. Nonetheless, the GSA method continues to be extensively utilized in diagnostic and research due to its relatively low cost and sample requirement.

Our study was focused on mitigating the disruptive influence of genomic waves—recurring wave-like patterns pervasive across the genome—which significantly impair the accuracy of detecting copy number variations (CNVs). By addressing these inherent challenges posed by genomic waves, our aim was to develop methodologies or techniques that improve the precision and reliability of CNV detection within genetic data analysis. To address this, we specifically employed k-means and k-NNs for clustering wave patterns and classifying samples, considering their simplicity and interpretability crucial when handling high-dimensional microarray data with hundreds of thousands of probes.

While these methods proved effective, the evolving realm of machine learning holds the potential for achieving even higher CNV detection accuracy in a neonatal setting. Recent advances in machine learning, particularly in deep learning and ensemble methods such as convolutional neural networks (CNN) and random forest, have demonstrated exceptional classification performance in medical imaging and molecular diagnosis applications [63,64]. CNNs, known as powerful tools for image recognition, could be instrumental in identifying CNVs in abnormal chromosomes by extracting key features like edges and specific banding patterns, which are crucial for detecting small chromosomal deletions and duplications. Nevertheless, the utilization of intricate machine learning models alongside high-quality data brings about trade-offs, including escalated costs, risks of overfitting, and challenges in interpretability, necessitating careful consideration in future research endeavors.

The utilization of machine learning methods extends far beyond the scope of this study, encompassing widely employed techniques such as k-means and k-NNs, which have significant applications in the diagnosis and exploration of diseases such as Autism Spectrum Disorder (ASD) [65,66]. These methods play a crucial role in analyzing complex datasets and aiding in the understanding and identification of patterns associated with ASD and other medical conditions. Furthermore, a variety of other machine learning approaches are utilized to analyze microarray data, highlighting the diverse array of methods employed in medical research [67,68]. 

Our approach involved employing customized machine learning models alongside the newly obtained mLRR values. Through validation, we demonstrated the capability to detect CNVs that remained undetected using existing LRR values, especially in detecting chromosomal disorders associated with CNVs. Furthermore, its accuracy was also confirmed through comparison with NGS data.

In typical microarray analysis, the log ratio is generally computed as the logarithm of the ratio of expression levels between two distinct samples. The concept of log ratios is extended and modified in the context of genotyping arrays, particularly when assessing copy number variations (CNVs). Notably, the log-R ratio demonstrates a correlation with gene expression levels [69]. Recent research determined gene expression levels by analyzing microarray images using log ratio [70,71]. 

We anticipate that our newly introduced mLRR value harbors extensive potential for versatile applications, extending its utility beyond genotyping to include the assessment of expression levels. This innovative metric holds promise for yielding more precise results compared to existing methods, thereby offering prospects for enhanced precision and comprehensive analyses.

Our study lacked continuous clinical observation to evaluate the long-term accuracy in predicting developmental disabilities among tested newborns. However, considering our success in minimizing the impact of genomic waves and obtaining accurate detection results, our method utilizing whole-genome SNP arrays could be considered one of the most effective approaches for screening chromosomal abnormalities in newborns. 

## 5. Conclusions

Comprehensive CNV screening using new methods has the potential to significantly improve the screening process for patients with developmental disabilities and congenital malformations due to rare mutations and CNV-related chromosomal disorders. Validation of CNV detection provides strong evidence of its effectiveness in identifying a wide range of genetic abnormalities inherited or newly acquired during pregnancy. Therefore, the newly developed genotyping analysis presented in this study shows promise as a routine clinical screening tool for newborns and individuals at high risk of genetic diseases.

## Figures and Tables

**Figure 1 diagnostics-14-00084-f001:**
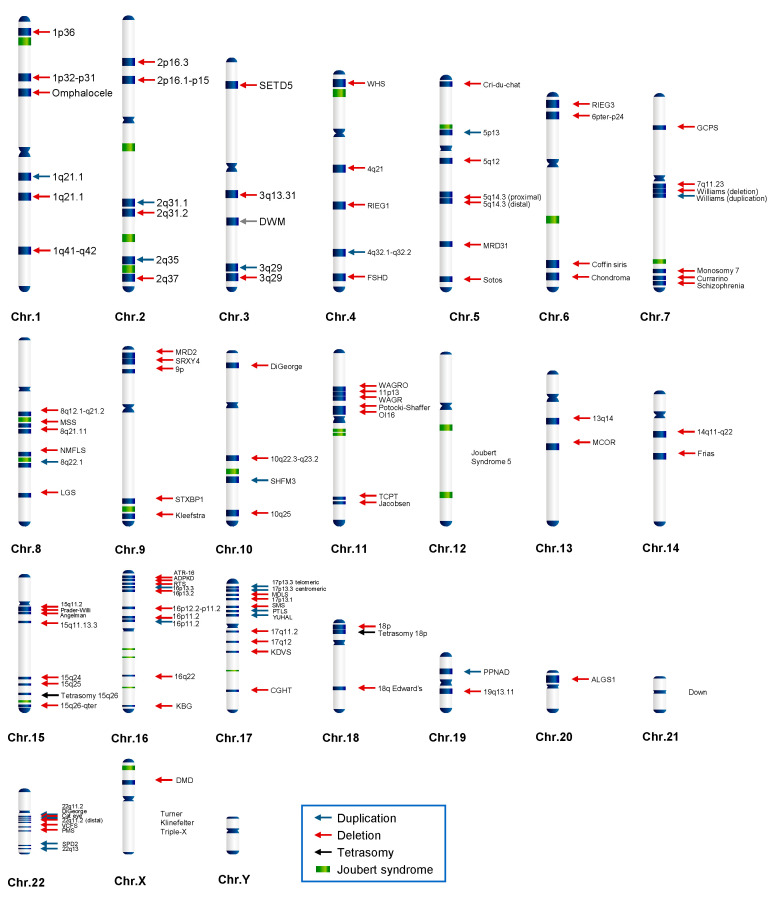
Schematic overview of 138 CNV-related chromosomal disorders. Red, blue, and black arrows indicate microdeletions, duplications, and tetrasomy, respectively. Green bars indicate Joubert syndrome types.

**Figure 2 diagnostics-14-00084-f002:**
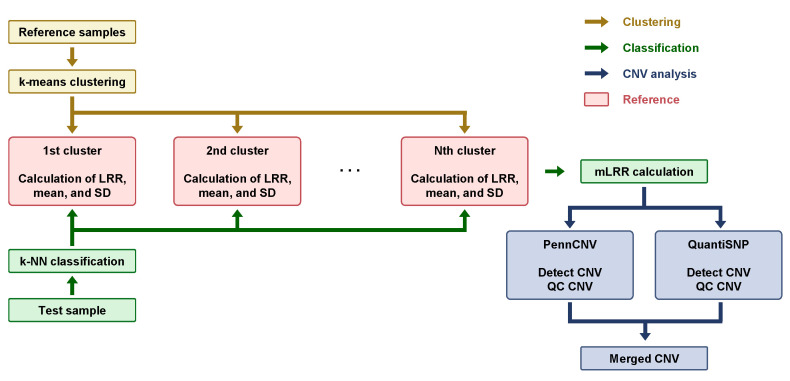
Key steps involved in Copy Number Variation (CNV) analysis using machine learning techniques. Reference samples indicate the clinical samples for clustering genomic waves. Golden brown arrows represent the clustering process, the green color indicates the classification process of the “Test sample”, and the navy color indicates the CNV analysis process after “mLRR calculation”.

**Figure 3 diagnostics-14-00084-f003:**
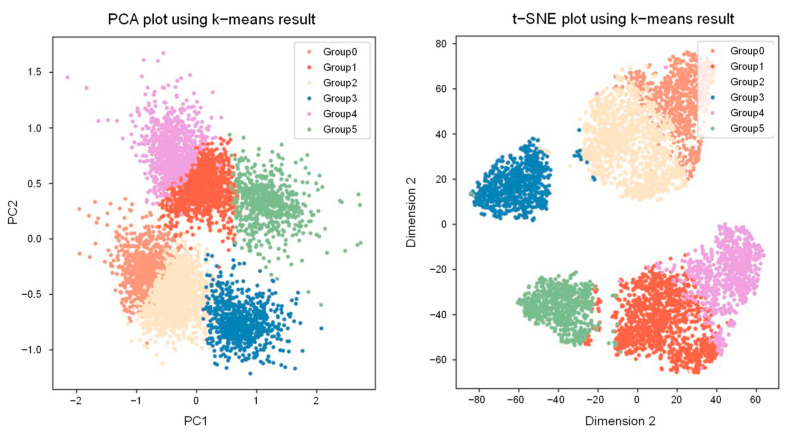
Principal component analysis and t-distributed stochastic neighbor embedding plots of 5399 clinical samples. Six colors represent each of the groups, which represent wave patterns.

**Figure 4 diagnostics-14-00084-f004:**
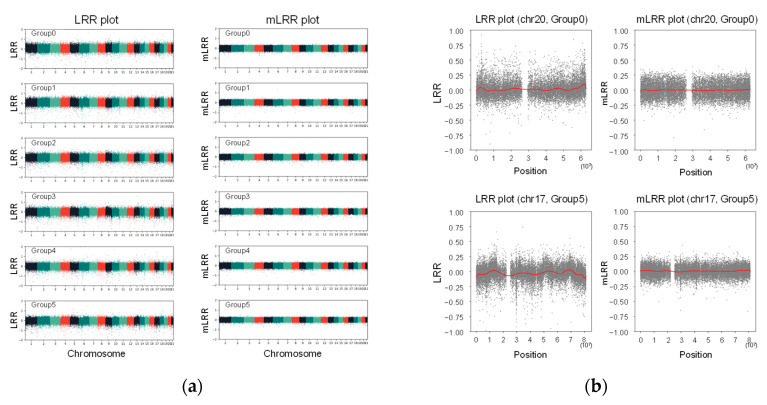
Comparison between Log R ratio (LRR) and modified LRR (mLRR) plots for each cluster. The horizontal axis of the plot represents the chromosome numbers, while the vertical axis represents the LRR and mLRR values: (**a**) Total results from the 21 chromosomes are represented. Each chromosome is identified by a different color; (**b**) Some examples of the end of the chromosomes are shown. The red lines represent the trend lines.

**Figure 5 diagnostics-14-00084-f005:**
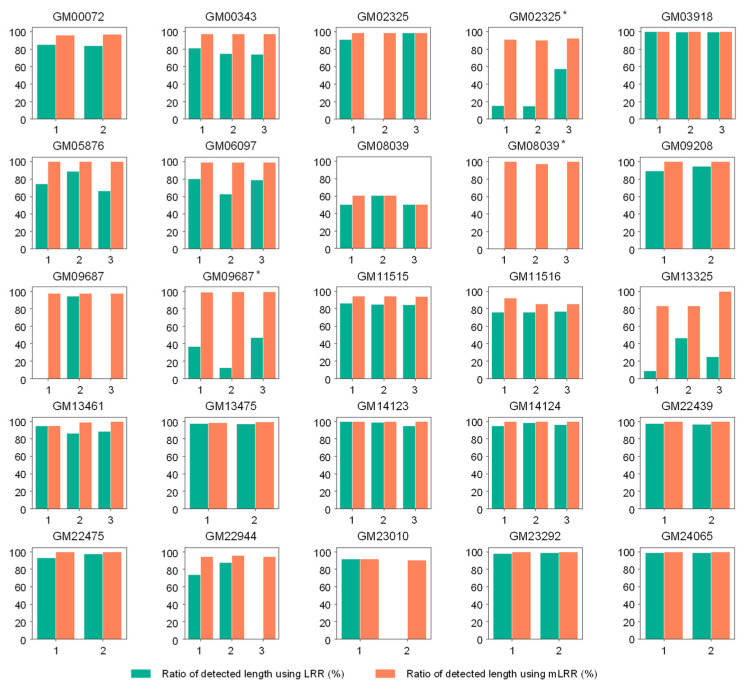
The results of CNV detection from Coriell cell line repository. The vertical axis represents the ratio of detected length to known CNV length. The horizontal axis indicates the number of repeats for each analysis. The green color indicates the results of the analysis using LRR, and the orange color represents the results of the analysis using mLRR. If the result is not detected, no bar is displayed. Asterisks (*) indicate cell lines that have two CNV regions. Please refer to Appendix A for more information.

**Figure 6 diagnostics-14-00084-f006:**
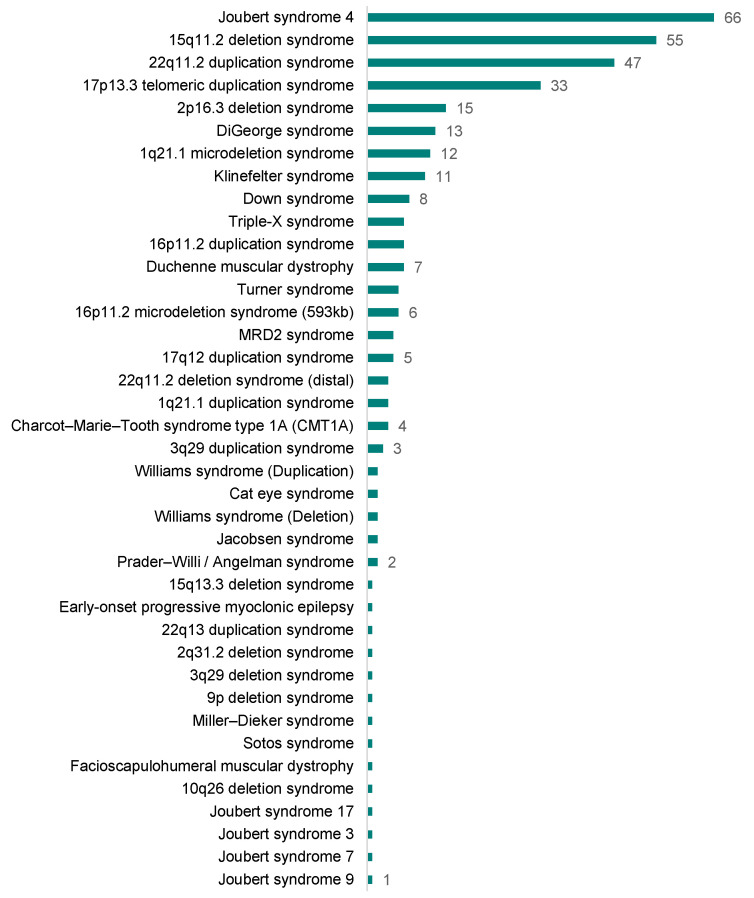
The number of identified chromosomal disorders from the screening of the 16,046 neonate samples. The numbers next to each bar represent the detected number, and bars of the same height represent the same number. Please refer to Appendix A for more information.

**Figure 7 diagnostics-14-00084-f007:**
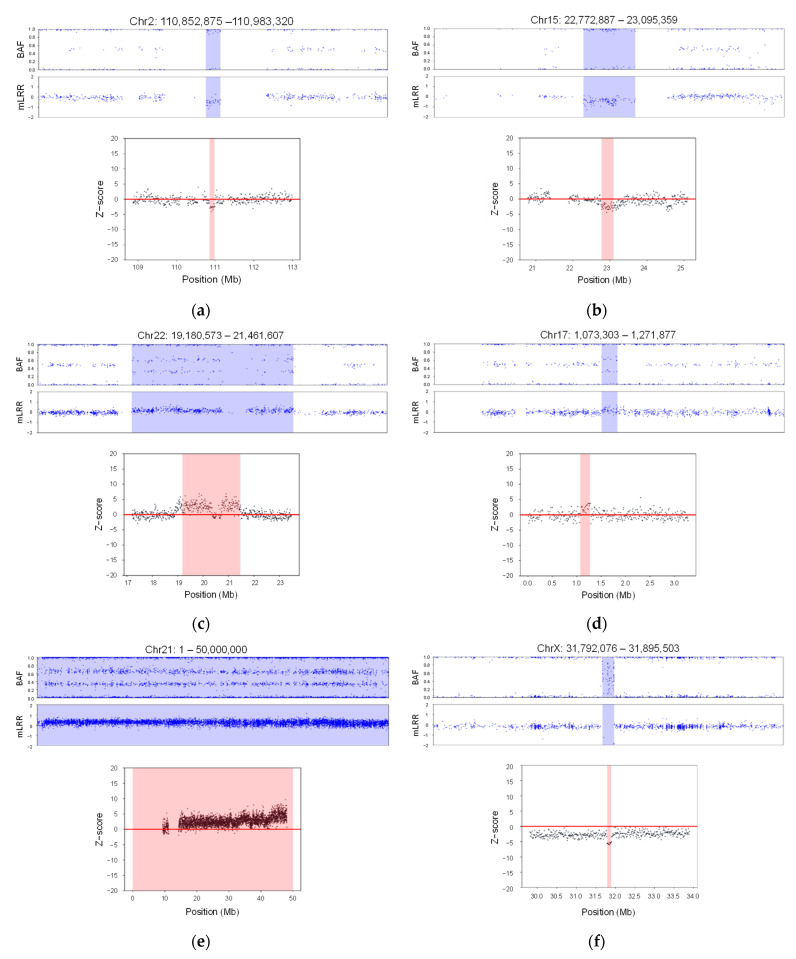
Validation results analyzing the same samples with a GSA array–based approach and NGS. A total of 8 chromosomal disorders are shown. The title displays the chromosome number along with the start and end positions of the detected region. The upper panel illustrates analysis results using the signal intensity patterns (B allele frequency, BAF) and modified log R ratio (mLRR). The vertical axis represents BAF and mLRR values, and each blue dot represents each value. The light blue color indicates the detected regions from the GSA. The lower panels represent the results from NGS analysis. The light pink regions represent the detected regions from the NGS. The vertical axis represents the z-score values, and the horizontal axis represents the positions: (**a**) Joubert syndrome 4; (**b**) 15q11.2 deletion syndrome; (**c**) 22q11.2 duplication syndrome; (**d**) 17q13.3 telomeric duplication syndrome; (**e**) Down syndrome; (**f**) Duchenne muscular dystrophy; (**g**) 1q21.1 deletion syndrome; (**h**) Williams syndrome (deletion).

## Data Availability

The datasets used and/or analyzed during the current study are available from the corresponding author upon reasonable request.

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
