# Peer review of "Improving CNV Detection Performance in Microarray Data Using a Machine Learning-Based Approach"

_diagnostics, 2023, doi:10.3390/diagnostics14010084_

Round 1
Reviewer 1 Report
Comments and Suggestions for Authors
The authors successfully employed k-means clustering procedures for improving CNV detection. Moreover, they proposed a modified LRR ratio which impacts the detection in a positive manner, compared with the existing approach. Nevertheless, in order to reach a larger target audience, the author should consider the following:
(i) in case of microarrays, log ratios are also computed in case of relative change in gene expression levels ([1] Microarray Image Analysis: From Image Processing Methods to Gene Expression Levels Estimation, IEEE Access, [2]An Automated cDNA Microarray Image Analysis for the determination of Gene Expression Ratios. IEEE ACM Trans. Comput. Biol. Bioinform. 2021,). I suggest that a discussion on how this new computation of log ratios can impact gene expression levels estimation can be introduced in a paragraf were the modified LRR computation is explained in detail.
(2) Moreover, considering the machine-learning approaches are found useful in all the research fields, a section within the discussions should be added where more complex machine-learning approaches that are found suitable (compared with K means) should be proposed as future work for the study in question. In other words, a discussion on machine learning on medical diagnosis should be added. (suggested bibliography:
[3]Evolution of Machine Learning in Tuberculosis Diagnosis: A Review of Deep Learning-Based Medical Applications. Electronics 2022, [4] Computational Complexity Case Study for Microarray Image Analysis Related to Machine Learning Approaches" Sensors 23, [5] DendrisChips® Technology for a Syndromic Approach of In Vitro Diagnosis: Application to the Respiratory Infectious Diseases. Diagnostics. 2018
)
Reviewer 2 Report
Comments and Suggestions for Authors
Goh et al. developed a new method using log R ratio for copy number variation (CNV) detection on microarray data. They analyzed a total of 16,046 Korean newborn samples and identified CNVs related to 39 genetic. They also validated the accuracy of the method using next generation sequencing (NGS) on a subset of the detected CNVs.
Chromosomal microarray analysis (CMA) has traditionally been the primary method for identifying copy number variations (CNVs) in clinical settings. However, recent studies (PMID: 30061371, PMID: 35085778) indicate a shift towards NGS, leaving limited room for the new development of CMA methods. Although the authors assert that CMA is more cost-efficient than Whole Genome Sequencing (WGS) using NGS for routine clinical testing, the authors neglected the truth that low-pass sequencing offer a better balance between cost and accuracy for CNV detection (PMID: 32344035, PMID: 32451733, PMID: 31447483, PMID: 37761262).
Major comments:
1. The utilization of K-means and k-nearest neighbor methods in this study is not novel for CMA analysis (PMID: 27479844, PMID: 23227143). It is essential to cite and compare with previous studies and highlight the novelty of the proposed method. Additionally, if the tool is publicly available, please provide information on its accessibility.
2. Is Figure 1 unique to this study and composed by the authors? If not, cite the original source.
3. Line 189 mentions the use of 36bp paired-end (PE) sequencing for validation. However, 36bp PE reads may be insufficient for high-quality alignment in challenging genomic regions where may CNVs reside, and this technique is considered outdated. Please consider employing a more advanced PE sequencing technique.
4. The manuscript lacks details on the analysis flow for paired-end NGS data.
5. Lines 122-123, “adjacent CNVs that were <200 kb apart were merged and filtered out based on the SNP number (>10), CNV length (>50 kb), and confidence score (>50)”. Clarify whether merging occurs when adjacent CNVs are of different types, such as deletions and duplications.
Minor comments:
1. Please spell out NGS for the first time it appears.
2. I wonder if this is a file format conversion issue. Otherwise, the resolutions of the figures need to be improved.
Round 2
Reviewer 2 Report
Comments and Suggestions for Authors
Thank you so much for the response. I'm curious about the reasons behind the unavailability of the tool for public access.
Author Response
Thank you for your interest and inquiry regarding the unavailability of our tool for public access. There are several reasons behind this decision. Firstly, our tool contains proprietary technology that we're safeguarding as part of our company's intellectual property. Additionally, maintaining a competitive advantage in our field is a priority, and therefore, we've chosen not to make the tool publicly available. I hope this explanation adequately addresses your query.